# An Image Encryption Scheme Based on Logistic Quantum Chaos

**DOI:** 10.3390/e24020251

**Published:** 2022-02-08

**Authors:** Yu Wang, Liquan Chen, Kunliang Yu, Yuan Gao, Yang Ma

**Affiliations:** 1School of Cyber Science and Engineering, Southeast University, Nanjing 210096, China; 230198994@seu.edu.cn (Y.W.); kunliang_yu@seu.edu.cn (K.Y.); 230209298@seu.edu.cn (Y.G.); mayang@jsca.gov.cn (Y.M.); 2Purple Mountain Laboratories, Nanjing 211100, China

**Keywords:** quantum computation, quantum logistic, quantum image processing, quantum image representation, image encryption

## Abstract

This paper proposes an image encryption scheme based on logistic quantum chaos. Firstly, we use compressive sensing algorithms to compress plaintext images and quantum logistic and Hadamard matrix to generate the measurement matrix. Secondly, the improved flexible representation of the quantum images (FRQI) encoding method is utilized for encoding the compressed image. The pixel value scrambling operation of the encoded image is realized by rotating the qubit around the axis. Finally, the quantum pixel is encoded into the pixel value in the classical computer, and the bit-level diffusion and scrambling are performed on it. Numerical analysis and simulation results show that our proposed scheme has the large keyspace and strong key sensitivity. The proposed scheme can also resist standard attack methods such as differential attacks and statistical analysis.

## 1. Introduction

Digital images are important information carriers and are widely used in various fields, such as social networks, education and medical systems [1,2]. Images can carry much redundant information, and it is easy to cause inadvertent privacy leakage in the transmission process, which eventually brings huge economic loss to users [3]. Especially in recent years, there have been many private information leakage incidents, which have aroused widespread discussion and concern among the public. Therefore, the research on image security is receiving more and more attention from scholars [1,4,5]. Image encryption can protect users’ private information from being leaked. However, traditional encryption methods such as AES, DES, 3DES, RC and BlowFish cannot protect image security because these algorithms cannot eliminate the characteristics of adjacent pixel correlation and information redundancy of ciphertext images [6]. Because the image has a high correlation between adjacent pixels and data redundancy, these characteristics lead to the unsatisfactory results of traditional encryption methods and are easy to be attacked [7,8]. Given the limitations of traditional encryption methods, many scholars have also proposed corresponding image encryption algorithms [9,10,11,12]. Image encryption algorithms based on chaotic systems have become a research hotspot.

The chaotic system is a nonlinear dynamic system and is difficult to predict. Meanwhile, it is sensitive to initial parameters and ergodic. Based on the above characteristics, Matthews first proposed an encryption algorithm based on chaos in 1989 [13]. Hua et al. proposed a high efficiency scrambling and diffusion image encryption algorithm for the problem of the privacy leaks of medical images [1]. The robustness of the algorithm is excellent, and the encrypted medical images can effectively resist image noise attacks. Liu et al. used Markov chains to improve the chaotic sequence distribution of the coupled lattice mapping (CML) [14], making the improved sequence distribution uniform. The improved sequence in this scheme can pass 15 test suites of NIST 800. However, the time complexity in this work increases with the introduction of nodes, and the efficiency may decrease exponentially. Goggin et al. used the recoil rotor model to measure the classical logistic system in 1990 and obtained the three-dimensional quantum logistic mapping [15]. Many scholars have proposed an image encryption algorithm based on quantum chaos based on Goggin’s work [16,17].

With the introduction of quantum computers [18], quantum image representation, as a critical information format, has also attracted widespread attention from researchers in recent years [16,19,20,21]. Quantum computers are pretty different from traditional computers in storage and calculation methods. In terms of calculation methods, taking 2D digital images as an example, the calculation of images by classic computers is matrix operations, and the calculation of images by quantum computers is the evolution of unitary operators. Therefore, many researchers have proposed corresponding algorithms to represent images on quantum computers [21,22,23,24]. Yao et al. proposed an algorithm to encode 2D classical images into pure quantum states [23]. They assumed that the quantum state’s probability amplitude represents the pixel value of the image on the quantum computer, and the quantum ground state represents the pixel’s position. In 2003, Venegas-Andraca et al. first proposed the image quantum encoding algorithm based on qubit lattice [19,25]. They assume that a qubit stores each pixel value of the input 2D image. Therefore, at least 2*n* bits are required to store the pixel value for the input image. However, this storage method does not realize the quantum acceleration function. In 2011, Le et al. designed a flexible representation of quantum images (FRQI) based on the qubit lattice theory [21]. They took the pixel value and position in the form of a quantum state tensor product. The qubits required in the encoding process are reduced from 2*n* bits to *n* bits, which improves storage efficiency. Zhang et al. also proposed the NEQR quantum image encoding method. Both NEQR and FRQI use the base state $|k\rangle$ to represent the pixel position, but it is worth noting that FRQI takes the pixel value during the encoding process [26]. The information is stored in the probability range, while NEQR stores the pixel value information in the ground state. Moreover, because NEQR’s quantum state is orthogonal and distinguishable in each ground state, the NEQR has a secondary acceleration in the quantum image preparation process, and the compression rate of the quantum image is 1.5 times higher than FRQI. The researchers propose the corresponding quantum image processing technology and machine learning algorithm for quantum images based on the quantum image representation algorithm. In [27], Ma et al. encode image information using qubits and implement image dilation and erosion operations in quantum computers. In Reference [28], the authors use competitive learning and quantum computing to develop a new classification algorithm that can classify inputs even when they are incomplete. Zhou et al. used novel enhanced quantum representation (NEQR) to encode images and performed Arnold transformation on the pixel values of quantum images to achieve pixel-level scrambling, but they did not eliminate the security threat of Arnold transformation [16].

Compressed sensing can break through the limitations of the Nyquist sampling theorem and use sparse data to restore the original signal [29,30,31,32]. Researchers have applied it to image encryption and machine learning [29,33]. Combining chaotic systems and compressed sensing theory to propose image encryption algorithms is a field worthy of in-depth study. In the image encryption process, the length of the encryption time largely depends on the data dimension. If the data dimension can be reduced, the encryption time of the encryption system can be decreased. At present, many researchers have proposed chaos theory and compressive sensing algorithms. For example, Chai et al. used 2D compressed sensing algorithms to encrypt images into cryptographic images with visual significance [29]. Wang et al. used a parallel compressed sensing algorithm to encrypt images, in encryption time and excellent encryption effect [4].

In order to ensure that the quantum image can be securely transmitted and used in the future quantum computer era, we convert the classical image into a quantum state and then encrypt it. This paper proposes an image encryption scheme based on logistic quantum chaos. The contributions of this paper are summarized as follows:We use quantum logistic mapping and the Hadamard matrix to generate the measurement matrix. This new matrix can improve encryption scheme security in the compressed sensing process in this paper.We use the FRQI quantum encoding method to perform quantum encoding on the compressed image, which can significantly reduce the amount of encoded data without affecting the encryption result.We combined the Bloch spherical surface to propose a new pixel value scrambling method. This new operation can reduce time complexity and improve system security in the encryption process.We use the improved FRQI to encode the classical image. In the new quantum image representation scheme, the pixels of the classical image are only related to one rotation angle of the Bloch sphere. On the premise that the other rotation angle is unchanged, it is easier to encrypt the quantum image, and the complexity of the new quantum representation scheme is the same as that of FRQI.

The structure of this paper is as follows: In Section 2, we give the corresponding essential work. In Section 3, we describe the encryption and decryption process of the proposed algorithm. Section 4 analyzes the image security after encryption, and Section 5 concludes this paper.

## 2. Related Work

### 2.1. Compressive Sensing (CS)

Suppose the N×1 dimensional signal *f* is a non-sparse signal [31]. We need to find an N×N dimensional orthogonal basis transform base matrix $\Psi_{N\times N}$ so that f has a sparse representation in the transform domain Ψ. Namely:(1)f=Ψx.

Therefore, the CS process can be expressed as:(2)y=ΦΨx=Φf=Ax.

The Φ is the measurement matrix. *A* represents the sensing matrix, which is an M×N dimensional matrix.

Since the dimension *M* of the measurement vector *y* is much lower than the dimension *N* of the observed signal *x*, (Equation 1) is an underdetermined system, making *y* impossible to accurately calculate the target signal *f*. However, CS theory proves that if the perception matrix *A* can meet the Restricted Isometry Property (RIP) condition [32], the original signal can be recovered from the measured signal.

Therefore, the measurement matrix needs to meet:(3)$$\left(1-\delta_{k}\right){{\left||x|\right|}^{2}}_{2}\leq {{\left||Ax|\right|}^{2}}_{2}\leq \left(1+\delta_{k}\right){{\left||x|\right|}^{2}}_{2}.$$
The $\delta_{k}$ represents isometry constant. Reconstruct the original problem accurately by transforming it into an $l_{1}$ norm problem.
(4)$$\min{\left||x|\right|}_{1}x.t.y=\Phi \Psi x.$$
The ${\left||x|\right|}_{1}$ represents the $l_{1}$ norm of the vector *x*.

The image reconstruction algorithm can use orthogonal matching pursuit (OMP) [4,33,34], matching pursuit (MP) [35], subspace tracking (ST) or basis tracking algorithm. In order to effectively restore the original image from the measured matrix, in this paper, we use the OMP method to reconstruct the compressed image.

### 2.2. Quantum Logistic Mapping

In 1990, Goggin [15,17] used the recoil rotor model to quantify the Logistic chaotic system. The Goggin method uses the coupling dissipative quantum system and the harmonic oscillator path to produce the new quantum system. Finally, Goggin obtained the 3D quantum modified logistic system corresponding to $\langle \delta a\delta a\rangle$ according to the Heisenberg dynamic motion equation δa [17]. The mathematical definition of the three-dimensional quantum logistic system is as follows:(5)$$\left\{\begin{array}{c} x_{n+1}=r\left(x_{n}-|x_{n}{|}^{2}\right)-ry_{n}\\ y_{n+1}=-y_{n}e^{-2\beta }+re^{-\beta }\left[\left(2-x_{n}-x_{n}^{*}\right)y_{n}-x_{n}z_{n}^{*}-x_{n}^{*}z_{n}\right]\\ z_{n+1}=-z_{n}e^{-2\beta }+re^{-\beta }[\left(2\left(1-x_{n}^{*}\right)z_{n}-2x_{n}y_{n}-x_{n}\right] \end{array}\right..$$

In (Equation 5), $x=\langle a\rangle ,\;y=\langle \delta a^{†}\delta a\rangle ,\;z=\langle \delta a\delta a\rangle$, δa describes the quantum fluctuations about 〈a〉 and δa→0. The *r* represents the control parameter, β is the dissipation parameter and ${x_{n}}^{*},{z_{n}}^{*}$ is the conjugate complex number of $x_{n},z_{n}$, and in the chaotic state when $x_{n}\in \left(0,1\right],\;y_{n}\in \left(0,0.1\right],\;z_{n}\in \left(0,0.2\right],\;\beta \in \left[6,+\infty \right],\;r\in 4$.

### 2.3. Quantum Encoding of Images

This section will use the quantum encoding method proposed in reference [22] to encode images and continue to encrypt images after encoding. The specific encoding method is as follows:(6)$$|I(\theta ,\varphi )\rangle =\frac{1}{2^{n}}\sum_{k=0}^{2^{2n}-1}|k\rangle |c_{k}\rangle .$$
(7)$$\left\{\begin{array}{c} |c_{k}\rangle =\cos\frac{\theta_{k}}{2}|0\rangle +e^{i\varphi_{k}}\sin\frac{\theta_{k}}{2}|1\rangle \\ \theta_{k}=\frac{\pi c_{k}}{255},\;0\leq \theta_{k}\leq \pi \\ \varphi_{k}=2\pi x_{k},\;0\leq \varphi_{k}\leq 2\pi \end{array}\right..$$

The θ and ϕ are the Bloch vector angles. The position of the *k*-th (*k* = 0, 1, …, $2^{2n-1}$) pixel is described as $|k\rangle$. The gray value is $|c_{k}\rangle$. The $x_{k}$ is a pseudo-random number in [0, 1]. In this article, we use the quantum logistic equation to generate $x_{k}$.

The quantum state of the encoded image satisfies the normalization condition [22]:(8)$$\left|||I\left(\theta ,\varphi \right)||\right|=\frac{1}{2^{n}}\sqrt{\sum_{k=0}^{2^{2n}-1}\left({\left(\cos\frac{\theta_{k}}{2}\right)}^{2}+{\left|e^{i\varphi_{k}}\right|}^{2}{\left(\sin\frac{\theta_{k}}{2}\right)}^{2}\right)}=1.$$

At the same time, reference [22] gives a fast implementation of image quantum coding, the specific description is as follows:

Step 1: Define quantum revolving door $R_{k}\left(\theta_{k},\varphi_{k}\right)$:(9)$$R_{k}\left(\theta_{k},\varphi_{k}\right)=\left(\begin{array}{cc} \cos\frac{\theta_{k}}{2} & -\sin\frac{\theta_{k}}{2}\\ e^{i\varphi_{k}}\sin\frac{\theta_{k}}{2} & e^{i\varphi_{k}}\cos\frac{\theta_{k}}{2} \end{array}\right).$$

Step 2: Define a controlled revolving door $CR_{k}\left(\theta_{k},\varphi_{k}\right)$:(10)$$CR_{k}\left(\theta_{k},\varphi_{k}\right)=\left(\sum_{j=0}^{2^{2n}-1}|j\rangle \langle j|\right)⊗I+|k\rangle \langle k|⊗R_{k}\left(\theta_{k},\varphi_{k}\right).$$

Step 3: Prepare the initial state of the qubit ${|0\rangle }^{⊗2n+1}$.

Step 4: To calculate the intermediate state, use the two-dimensional Hadamard matrix and the identity matrix $|H\rangle$.
(11)$$|H\rangle =\left({\left(\frac{1}{\sqrt{2}}\left(\begin{array}{cc} 1 & 1\\ 1 & -1 \end{array}\right)\right)}^{⊗2n}⊗\left(\begin{array}{cc} 1 & 0\\ 0 & 1 \end{array}\right)\right){|0\rangle }^{⊗2n+1}=\frac{1}{2^{n}}\left(\sum_{k=0}^{2^{2n}-1}|k\rangle \right)|0\rangle .$$

Step 5: Perform $CR_{k}\left(\theta_{k},\varphi_{k}\right)$ on the intermediate state:(12)$$CR_{k}\left(\theta_{k},\varphi_{k}\right)|H\rangle =\frac{1}{2^{n}}\left[\left(\sum_{j=0,j\neq k}^{2^{2n}-1}|j\rangle \right)⊗|0\rangle +|k\rangle ⊗|c_{k}\rangle \right].$$

Step 6: In order to obtain the encoded quantum image, perform $CR_{k}\left(\theta_{k},\varphi_{k}\right)$ continuously on the intermediate state:(13)$$I\left(\theta ,\varphi \right)=\left(\prod_{k=0}^{2^{2n}-1}CR_{k}\left(\theta_{k},\varphi_{k}\right)\right)|H\rangle .$$

## 3. Algorithm Description

In this section, we select Lena.bmp as plaintext, in which *M* = 256 and *N* = 256, where *M* and *N* are the length and width of plaintext, respectively. Quantum logistic function’s parameters are: *x* = 0.5, *y* = 0.0032, *z* = 0.12, *r* = 3.99, β = 7. The iterative number of quantum logistic is related to the size of plaintext images. Orthogonal matching pursuit (OMP) algorithm is used to reconstruct the image after compressive sensing.

### 3.1. Encryption Process

#### 3.1.1. Compressed Sensing Process

In the encryption process, the plaintext image is represented by *I*.

Step 1: The two-dimensional wavelet transforms to the plaintext image *I*, transformation model is haar transform model. After the two-dimensional wavelet transforms, we can obtain the decomposition coefficients c and length s for each layer.
(14)$$\left\{\begin{array}{c} \left[c,s\right]=wavedec2\left(I,N{,}^{\prime }haar^{\prime }\right)\\ s=[A(N)|H(N)|V(N)|D(N)|H(N-1)|\ldots |H(1)|V(1)|D(1)] \end{array}\right..$$
*N* is the layer of the two-dimensional wavelet transform, and we set *N* = 4 in step 1. *H*, *V* and *D* are three directions of plaintext image. The *A* and *H* are low frequency and the high frequency of each layer, respectively. The low frequency contains the feature of the plaintext image, and the high frequency contains the details of the plaintext image. To ensure the quality of the reconstructed image, we only compress the high-frequency data, and the low-frequency data of the plaintext image will not be compressed.

Step 1.1: Use QR decomposition to low-frequency data *cA*,
(15)$$\left\{\begin{array}{c} cA=c(A(N))\\ \left[q,r]=QR(cA\right) \end{array}\right..$$
*N* is the two-dimensional wavelet transform layer number and *N* = 1, 2, 3, 4. Both *q* and *r* are matrices.

Step 1.2: Extract the diagonal data in the *r* and append it to the end of matrix *q* as one of the decryption keys. The matrix *r* after being extracted data replaces the *cA* as a part of the plaintext image to encrypt.
(16)$$\left\{\begin{array}{c} temp(i)=r(i,i)\\ r(i,i)=0\\ q(:,end+1)=temp \end{array}\right.$$
The vector *temp* is a temporary vector, and *i* is the position index of the matrix *r*.

Step 1.3: Grouping high-frequency data:(17)$$\left\{\begin{array}{c} cB_{h}\left(i\right)=c\left(H\left(i\right)\right)\\ cB_{v}\left(i\right)=c\left(V\left(i\right)\right)\\ cB_{d}\left(i\right)=c\left(D\left(i\right)\right) \end{array},\;i=1,2,3,\ldots ,N\right..$$

Equation (Equation 17) can be used to obtain high-frequency data of each layer *cB*, and then *cB* is transformed to the matrix according to the length of the decomposition coefficient *s* for each layer. Namely,
(18)$$cB_{h,v,d}\left(i\right)=reshape\left(cB_{h,v,d}\left(i\right),s_{h,v,d}\right).$$

Step 2: To ensure the high reconstruction quality of the plaintext image, the high-frequency data after wavelet decomposition needs to be sparse operation. We use Algorithm 1 to sparse the high-frequency data matrix.
**Algorithm 1** Calculating the sparsity threshold**Input:**$cB=cB_{h,v,d+r}$.**Output:** sparsity matrix $STD_{i}$. 1: Set sparse rate: 0.10≤iSTD≤0.15; 2: Obtain length and width of the matrix: $s_{h,v,d}$. 3: Sparsify each row of the matrix,  std = 0; // sparsity threshold  iSTD = sparsity(cB, std); // sparsity rate 4: while std < 0.1 and 0.10 ≤ *iSTD* ≤ 0.15  std = std + 1 × 10^−4^  iSTD = sparsity(cB, std);  end 5: pos = find(abs(cB)< std); 6: cB(pos) = 0;

Step 3: After sparsification, the overall sparsity may reach the *iSTD*, but some data may be concentrated within a particular region. Therefore, it is necessary to displace the data and distribute these pixel points evenly throughout the sparse matrix. Use Equation (Equation 19) to confuse the matrix,
(19)$$\left[STD_{i},index\right]=josephTraverse\left(STD_{i},start,space\right).$$
Both *start* and *space* are the confusing start and end positions, respectively. The *index* represents the position of the original matrix, which is a part of the decryption key. The *josephTraverse* is the confusion function.

Step 4: Building the measurement matrix Φ.

Repeat Algorithm 2 until each layer’s high-frequency and low-frequency data have corresponding Hadamard measurement matrices $\Phi_{M\times N/2^{i}}^{ij}$.

Step 5: Use Equations (Equation 20) and (Equation 21) to compressive high-frequency and low-frequency data,
(20)$$y_{ij}=\Phi_{s(i1)\times s(i,2)/2}^{ij}STD_{ij},i\in \left(1,N\right),j\in \left(1,3\right),$$
(21)$$r=\Phi_{s(1,1)\times s(1,2)/2}^{1,1}r.$$
Splice all matrices into the compressed matrix *x* after compressive sensing.

Step 6: Quantizing the compressed matrix *x* can facilitate bit-level encryption and quantum pixel conversion operations.
(22)$$x=uint8(\frac{x-\min(x)}{\max(x)-\min(x)}).$$

**Algorithm 2** Building Hadamard measurement matrix based on quantum logistic system**Input:** sampling rate: sample = 0.5 and parameters of the quantum logistic system.**Output:** Measurement matrix Φ. 1: Obtain size of STDi, marked as [m,n].  [m,n] = size(STDi); 2: The Hadamard matrix H was calculated according to the sample rate.  H = Hadamard(m, n × sample); 3: The iterative quantum logistic mapping generates the confusion vector cs. To ensure  the security of the confusion vector, we discard the data of the first 200 iterations and   take the values from 201 to 201 + (m × n/2) as the confusion vector of the Hadamard   matrix. The values are transposed into a matrix and denoted as cs’.  cs(201: 201 + (m × n/2)) = quantum_logistic(parameters);  cs’ = reshape(cs, m, n/2); 4: To sort row and col of confusion matrix cs’, denote the index after sorted as index.  while i=row and col  [:, index] = sort(cs’ (i));  cs’(i) = index;  end 5: Confuse the Hadamard matrix,
(23)Φ=Hadamard(index);


#### 3.1.2. Quantum Pixel-Level Encryption

Step 7: Use Equations (Equation 9)–(Equation 13) to quantum code the image *x* after compressive sensing, the size of *x* is *M*/2 × *N* and the coding result is recorded as I(θ,φ).

Step 8: Scrambling operations in quantum pixel level. From Section 2.3, we know that each pixel value is only related to $\theta_{k}$ and not related to φ. Therefore, we can let φ remain unchanged and rotate $\theta_{k}$ to change the gray value.

Calculate the rotation angle $\Delta \theta_{k}$:(24)$$\theta_{1}=\;\mathrm{mod}\;\left(abs\left(\frac{q_{2}}{q_{3}}\times 2^{13}\right),1\right)\times \pi ,$$
(25)$$\Delta \theta_{k}=\theta_{k}-\theta_{1k}$$

$q_{1},q_{2},q_{3}$ are pseudo-random sequences generated by the quantum logistic system. $q_{1}$ is involved in the quantum encoding generation of images.

Let the Bloch sphere coordinates of $|c_{k}\rangle$ be $\left(x_{k},y_{k},z_{k}\right)$ to make $|c_{k}\rangle$ rotate toward the Bloch sphere to the point (0, 0, −1). The rotation axis should be:(26)$$R_{k}=\frac{\left(x_{k},y_{k},z_{k}\right)\times \left(0,0,-1\right)}{|\left(x_{k},y_{k},z_{k}\right)\times \left(0,0,-1\right)|}=\frac{\left(-y_{k},x_{k},0\right)}{\sqrt{x_{k}^{2}+y_{k}^{2}}}.$$

The rotation matrix is,
(27)$$M_{k}=\cos\frac{\Delta \theta_{k}}{2}I-i\sin\frac{\Delta \theta_{k}}{2}\left(R_{k}\times \sigma \right).$$

*I* is the unit matrix, and *i* is the imaginary unit. σ represents the Bubbleley matrix $\sigma =(\sigma_{x},\sigma_{y},\sigma_{z})$:(28)$$\sigma_{x}=\left(\begin{array}{cc} 0 & 1\\ 1 & 0 \end{array}\right),\sigma_{y}=\left(\begin{array}{cc} 0 & -i\\ i & 0 \end{array}\right),\sigma_{z}=\left(\begin{array}{cc} 1 & 0\\ 0 & -1 \end{array}\right).$$

Controlled revolving doors:(29)$$CM_{k}=\left(\sum_{j=0,j\neq k}^{2^{2n}-1}|j\rangle \langle j|\right)⊗I+|k\rangle \langle k|⊗M_{k}.$$

The rotation on the sphere can be defined as:(30)$$|I^{\prime }\left(\theta ,\varphi \right)\rangle =\left(\prod_{k=0}^{2^{2n}-1}CM_{k}\right)I|\left(\theta ,\varphi \right)\rangle =\frac{1}{2^{n}}\sum_{k=0}^{2^{2n}-1}|k\rangle |{c^{\prime }}_{k}\rangle .$$

#### 3.1.3. Bit-Level Encryption

Step 9: Quantum encoding image is scrambled in bit-level. Calculate the pixel value $c^{\prime }$ according to $\Delta \theta_{k}$. Use Equation (Equation 25) to scramble pixel value $c^{\prime }$.
(31)$$\left\{\begin{array}{c} X=quantum\text{\_}Logistic(M\times N\times 8,paramters)\\ \begin{array}{c} Q=dec2hex(\left(X\times 2^{13}\right)\;\mathrm{mod}\;256)\\ C_{1}=c_{1}^{\prime }⊕Q_{1}⊕c_{M\times N}^{\prime }\\ C_{i}=c_{i}^{\prime }⊕Q_{i}⊕c_{i-1}^{\prime },i\in \left(2,M\times N\right) \end{array} \end{array}\right..$$

*X* is the pseudo-random value generated by the iterative quantum logistic system. The ⊕ is the bitwise-XOR operation.

Step 10: Semi-ciphertext *C* is scrambled by the josephTraverse function in bit level.
(32)[C,index]=josephTraverse(C,start,space)
At this point, the encryption process is all over. The encryption process flowchart is shown in Figure 1. Figure 2, Figure 3 and Figure 4 show the encryption and decryption results of the proposed scheme.

### 3.2. Decryption Process

The decryption process is the reverse process of the encryption process. The details are as follows:

Step 1: Use the *ijosephTraverse* function to inverse scrambling on the ciphertext *C*.
(33)$$C^{\prime }=ijosephTraverse\left(C,index\right).$$
The *index* is the coordinate before *josephTraverse*. *C′* is the decryption result.

Step 2: Bit-level inverse diffusion.
(34)$$\left\{\begin{array}{c} X=quantum\text{\_}Logistic(M\times N\times 8,paramters)\\ \begin{array}{c} Q=dec2hex(\left(X\times 2^{13}\right)\;\mathrm{mod}\;256)\\ c_{i}=C_{i}^{\prime }⊕Q_{i}⊕c_{i-1}^{\prime },i\in \left(2,M\times N\right)\\ c_{1}=C_{1}^{\prime }⊕Q_{1}⊕c_{M\times N}^{\prime } \end{array} \end{array}\right..$$

The *X* is the pseudo-random value generated by the iterative quantum logistic system. The ⊕ is the bitwise-XOR operation.

Step 3: Decrypt the quantum coding image. The rotation matrices used in the encryption process are all unitary matrices and conform to the basic principles of quantum mechanics [21]. Therefore, we only need to change rotation matrices to conjugate transpose to decrypt the ciphertext in the decryption process.

$M_{k}$ is rotation matrices. Let $M_{k}^{+}$ be the conjugate transpose matrix of $M_{k}$. According to unity, we can obtain $M_{k}^{+}$:(35)$$M_{k}M_{k}^{+}=M_{k}^{+}M_{k}=I.$$

Calculate the decryption-controlled revolving door $C_{k,1}$ according to Equation (Equation 29). Use Equation (Equation 36) to decrypt quantum coding ciphertext image:(36)$$|I(\theta ,\varphi )\rangle =\left(\prod_{k=0}^{2^{2n}-1}C_{k,1}\right)|I^{\prime }\left(\theta ,\varphi \right)\rangle .$$

Before decrypting at the bit level, the decrypted ciphertext image $|I(\theta ,\varphi )\rangle$ needs to be converted into a gray value by quantum measurement. In order to achieve quantum measurement, we define the measurement symbol *K*,
(37)$$K=\sum_{k=0}^{2^{2n}-1}\left(k_{0}^{\left(n\right)}p_{0}^{\left(n\right)}+k_{1}^{\left(n\right)}p_{1}^{\left(n\right)}\right).$$
$p_{0}^{\left(n\right)}=|n\rangle |0\rangle \langle n|\langle 0|$ and $p_{1}^{\left(n\right)}=|n\rangle |1\rangle \langle n|\langle 1|$ are a set of orthogonal projection matrices corresponding to the eigenvalues $k_{0}^{\left(n\right)}$ and $k_{1}^{\left(n\right)}$ of *K*.

For the decrypted quantum image $|I(\theta ,\varphi )\rangle$, the probabilities of obtaining eigenvalues $k_{0}^{\left(n\right)}$ and $k_{1}^{\left(n\right)}$ with the measurement symbol *K* are:(38)$$\left\{\begin{array}{c} P\left(k_{0}^{\left(n\right)}\right)=\langle I(\theta ,\varphi )|k_{0}^{\left(n\right)}|I(\theta ,\varphi )\rangle =\frac{1}{2^{2n}}\cos^{2}\frac{\theta_{n}}{2}\\ P\left(k_{1}^{\left(n\right)}\right)=\langle I(\theta ,\varphi )|k_{1}^{\left(n\right)}|I(\theta ,\varphi )\rangle =\frac{1}{2^{2n}}{\left|e^{i\varphi_{n}}\right|}^{2}\sin^{2}\frac{\theta_{n}}{2} \end{array}\right..$$

The gray value of the *n*-th pixel $c_{n}$:(39)$$\left\{\begin{array}{c} \theta_{n}=2\arcsin\left(\sqrt{2^{2n}P\left(m_{1}^{\left(n\right)}\right)}\right)\\ c_{n}=ent\left(\frac{\left(2^{8}-1\right)\theta_{n}}{\pi }\right) \end{array}\right..$$

Step 4: Inversely scramble the image $c_{n}$.

Step 5: Use OMP to reconstruct the image of the decrypted compressive sensing semi-plaintext image.

Step 6: Use inverse wavelet transform to restore the plaintext image *P*.

The decryption process flowchart is shown in Figure 5.

## 4. Security Analysis

### 4.1. Keyspace Analysis

An excellent image encryption algorithm should have a large enough keyspace to resist the brute force attack analysis [36]. In theory, the keyspace needs to reach the power of $2^{100}$ to resist the brute force attack analysis of the attacker [37]. In our proposed algorithm, our algorithm key consists of two parts:Quantum logistic parameters: x,y,z,r,β and the number of max iterations; the number of wavelets transforms layers *N*, the start position and interval distance of the *josephTraverse* algorithm; the quantum logistic parameter in the quantum encoding process and the rotation angle θ.The *q* matrix and the *josephTraverse* generated the *index* during the encryption process.

Assuming that the calculation accuracy of the computer is 10–14, our algorithm keyspace is: ${\left(10^{14}\right)}^{16}\times {\left(10^{14}\right)}^{2(MN)}=\left(10^{224}\right)\times {\left(10^{14}\right)}^{2(MN)}>2^{300}.$

This result is much larger than the standard requirement of $2^{100}$, so the encryption algorithm we propose can resist brute force attack analysis. As shown in Table 1, compared with the research in references [4,17,33,38,39,40], our image encryption algorithm has a large enough keyspace to withstand all types of brute force attacks.

### 4.2. Key Sensitivity Analysis

In Figure 6, we use Lena256 as the encryption object to test the key sensitivity of our encryption system. During the test, we used the original key as: Key=[x=0.5, y=0.0032, z=0.12, r=3.99, β=7, q, index]. We add $10^{-10}$ to the above keys respectively and use the modified key to decrypt the ciphertext in the decryption process. The decryption result is shown in Figure 6. From Figure 6, we can see that even if the key changes $10^{-10}$, the wrong key cannot decrypt the ciphertext correctly, proving our encryption scheme is highly sensitive to the key.

### 4.3. Statistical Analysis

#### 4.3.1. Histogram Analysis

Histogram analysis is one of the essential indicators to measure the security of an encryption algorithm. The histogram can measure the degree of pixel change of a ciphertext image. An excellent encryption algorithm can eliminate the correlation between adjacent pixels in the picture and the histogram feature of the original plaintext information. Section 4.3 uses Lena, the Boat, and 7.2.01.tiff as the plaintext image. The (a), (c) and (e) of Figure 7 show that the plaintext histogram has uneven peaks, reflecting the pixel values distribution of the original images. From (b), (d) and (f) of Figure 7, it can be found that the histograms of the dense image are different from the histogram of the original images, and the uneven peaks become uniformly distributed. The attacker may not obtain the corresponding plaintext information by analyzing the histogram of the ciphertext image. Therefore, the proposed image compression and encryption algorithm can resist statistical analysis attacks.

#### 4.3.2. Adjacent Pixel Correlation

There is a strong correlation between adjacent pixels of a plaintext image, and an attacker can crack the algorithm by analyzing the correlation between adjacent pixels. Therefore, the correlation between pixels needs to be reduced to avoid statistical attacks in the design of the encryption algorithm. In this article, we use Equation (Equation 40) to calculate the correlation between adjacent pixels:(40)$$r_{xy}=\frac{\mathrm{cov}(x,y)}{\sqrt{D(x)}\sqrt{D(y)}}$$
In Equation (Equation 40), $\mathrm{cov}\left(x,y\right)=\frac{1}{N}\sum_{i=1}^{N}\left(x_{i}-E\left(x\right)\right)\left(y_{i}-E\left(y\right)\right),D\left(x\right)=\frac{1}{N}\sum_{i+1}^{N}{\left(x_{i}-E\left(x\right)\right)}^{2}$, $E\left(x\right)=\frac{1}{N}\sum_{i=1}^{N}x_{i}$. *x* and *y* are adjacent pixel values. *N* is the total number of pixels selected from the image.

We select 17 images of different sizes as plaintext for encryption. Randomly select 10,000 pairs of adjacent pixel values, and calculate its correlation in three different directions: vertical, horizontal and diagonal. The specific calculation results are shown in Table 2, and the corresponding adjacent pixel correlation distribution is shown in Figure 8. Finally, we choose Lena’s ciphertext image to show the horizontal, vertical and diagonal pixel distribution in Figure 9.

We can see that all the pixel values in the plaintext Lena is concentrated near a certain line, (a), (c) and (e) of Figure 9. However, in the pixel distribution of the ciphertext, the pixel values are evenly distributed throughout the space in (b), (d) and (f) of Figure 9, proving that the proposed encryption algorithm can resist statistical attack analysis.

At the same time, we also compare our proposed algorithm with other similar encryption algorithms using an image of Lena256. Table 3 shows that our algorithm correlation coefficient of adjacent pixels is better than similar encryption algorithms.

#### 4.3.3. Information Entropy

In 1948, Shannon proposed [41] the concept of information entropy, which is the degree of confusion in the information source. The concept of information entropy can describe the uncertainty of the ciphertext. In an ideal state, the closer the information entropy is to 8, the better the encryption effect.
(41)$$H\left(s\right)=\sum_{i=0}^{2^{L}-1}p(s_{i})\log_{2}\frac{1}{p(s_{i})}$$

We calculate the information entropy of 17 ciphertext images of different sizes, shown in Table 4. From Table 4, we can see that the information entropy of the 17 ciphertext images is close to the ideal value of 8. These results indicate that the ciphertext images are very messy and can resist the statistical attack analysis of the attacker.

Table 4 uses Lena256 as the information entropy test object to compare with other similar encryption algorithms. From the comparison results in Table 5, the information entropy of our proposed algorithm is better than similar algorithms.

### 4.4. Robustness Analysis

Robustness is an important indicator that shows whether a cryptographic system can resist interference. In Section 4.4, we use noise attacks and blocking attacks to verify the robust performance of our proposed cryptographic system.

#### 4.4.1. Noise Attack Analysis

Common noise types include Gaussian noise, Poisson noise, multiplicative noise and salt and pepper noise. Salt and pepper noise has the most significant impact on the ciphertext image among all the noises. We add salt and pepper noise to the ciphertext image in Section 4.4.1 and decrypt the ciphertext after adding noise. Figure 10 shows the decrypted image with different noise intensities. From Figure 10, we can see that when the salt and pepper noise intensity reaches 0.1, the decryption result can also be distinguished with the naked eye, proving that our algorithm can resist noise attacks.

#### 4.4.2. Crop Attack Analysis

We crop the 8 × 8, 16 × 16 and 32 × 32 information of the ciphertext image for testing. The quality of the decrypted image will decrease as the block size increases. In Figure 11, Figure 12 and Figure 13, we show the ciphertext image cropped with different sizes of block and the corresponding decrypted image. From the experimental results, it can be seen that when the crop size is 32 × 32, the decryption result has the greatest impact, but even when the decryption result is the worst, we can still understand the information carried by the decrypted image, which proves that our algorithm, to a certain extent, can withstand tailoring attacks.

### 4.5. Resistance to Difference Analysis

Differential attack refers to a cryptanalysis technique that obtains the encryption key by analyzing the difference between two ciphertexts after making small changes on the plaintext. The number of pixels changes rate (NPCR) and unified average changing intensity (NACI) are generally used as evaluation indicators when measuring whether a cryptographic system can resist differential attacks. The ideal values of the two indicators are NPCR ≥ 99.6093% and UACI ≥ 33.4635%.
(42)$$\mathrm{NPCR}=\frac{\sum_{i,j}D(i,j)}{W\times H}\times 100$$
(43)$$\mathrm{UACI}=\frac{1}{W\times H}\left[\sum_{i,j}\frac{\left|c_{1}\left(i,j\right)-c_{2}\left(i,j\right)\right|}{255}\right]\times 100$$
(44)$$D\left(i,j\right)=\left\{\begin{array}{cc} 1 & ifc_{1}\left(i,j\right)\neq c_{2}\left(i,j\right)\\ 0 & ifc_{1}\left(i,j\right)=c_{2}\left(i,j\right) \end{array}\right.$$

*W* and *H* represent the image’s width and height, respectively. $c_{1}$ and $c_{2}$ are two ciphertext pixels of images. *i* and *j* are the same position index of the $c_{1}$ and $c_{2}$.

We calculated the NPCR and UACI of 15 ciphertext images with 1bit data changed in Table 6. We can see that our NPCR and UACI are close to their respective ideal values, proving that our algorithm can resist differential attacks. Table 7 uses Lena256 as the NPCR and UACI test object to compare with other similar encryption algorithms. From the comparison results, the NPCR and UACI of our proposed algorithm are better than similar algorithms.

### 4.6. Time Complexity Analysis

Time complexity analysis is a critical indicator to measure an encryption algorithm [42]. The primary encryption time consumption concentrates on wavelet transform (Step 1), sparse (Step 2), iterative quantum logistic system (Steps 2–5), normalization (Step 6), josephTraverse (Steps 3, 5), image quantum coding (Step 7) and quantum scrambling (Step 8) in the encryption process. In the compressive sensing process (wavelet transform (Step 1) and sparse (Step 2)), the time complexity of constructing the measurement matrix for compressed sensing is O(nlogn). In the quantum encoding stage, we use the same number of quantum gates in the encoding method as FRQI, so the computational complexity of the two is the same, that is, $O(\log^{2}\left(n\right))$, and the time complexity during the axis rotation phase is easy to calculate as $O(n^{2})$. The total time complexity of bit-level scrambling and diffusion does not exceed $O(n^{2})$ in Steps 2–5, Step 6 and Steps 3 and 5. The proposed encryption algorithm has better performance than classical algorithms in computational complexity.

## 5. Conclusions

This paper uses the FRQI quantum coding method on the classical computer. Firstly, we use compressive sensing algorithms to compress plaintext images to reduce data dimensions for shortening the time of quantum encoding. Secondly, FRQI is used to encode the compressed image and the scrambling operation by rotating the qubit around the axis. Finally, the bit-level diffusion and scrambling in the classical computer is performed. The computational complexity is not high in the whole encryption process. However, there is no way to achieve quantum parallelism in the simulation of classical computers, resulting in low computational efficiency. Although we compress the image data in the encryption process, the encryption time is still unsatisfactory. Because the existing quantum image coding methods and image storage models are currently in the theoretical research stage, the coding efficiency is not satisfactory. In addition, numerical simulation and theoretical analysis show that our proposed algorithm has better performance when compared with similar algorithms. Therefore, our scheme can guarantee the security of the image during transmission in the future quantum era.

## Figures and Tables

**Figure 1 entropy-24-00251-f001:**
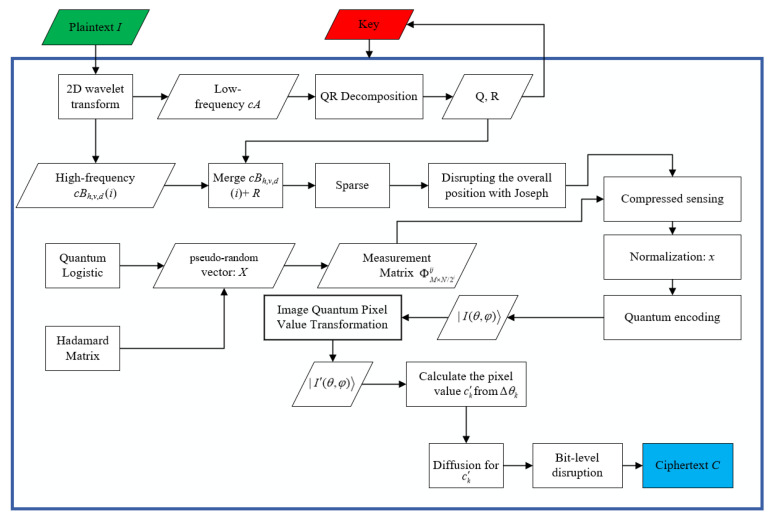
Encryption flowchart.

**Figure 2 entropy-24-00251-f002:**
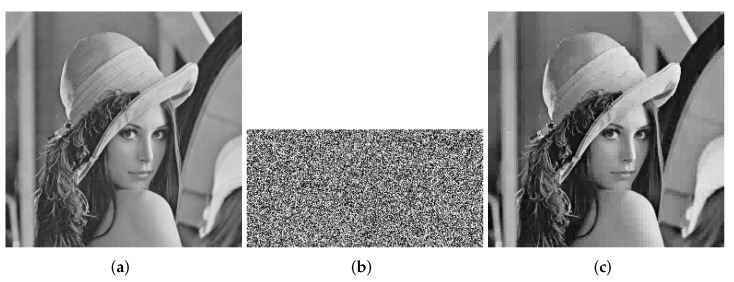
Lena (256 × 256). (**a**) Plaintext image of Lena. (**b**) Ciphertext image of Lena. (**c**) Decrypted Lena.

**Figure 3 entropy-24-00251-f003:**
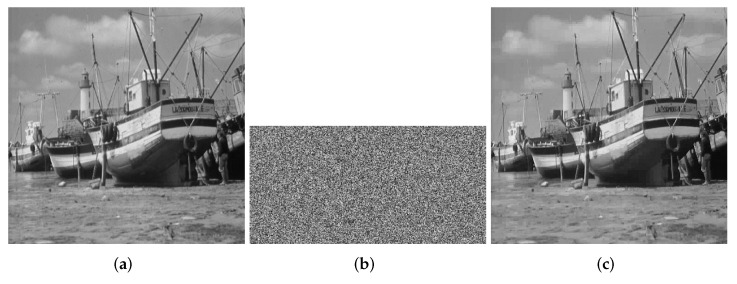
boat.tiff (512 × 512). (**a**) Plaintext image of boat. (**b**) Ciphertext image of boat. (**c**) Decrypted boat.

**Figure 4 entropy-24-00251-f004:**
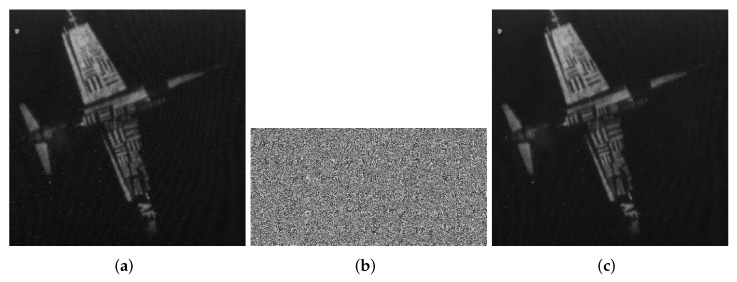
7.2.01.tiff (1024 × 1024). (**a**) Plaintext image of 7.2.01. (**b**) Ciphertext image of 7.2.01. (**c**) Decrypted 7.2.01.

**Figure 5 entropy-24-00251-f005:**
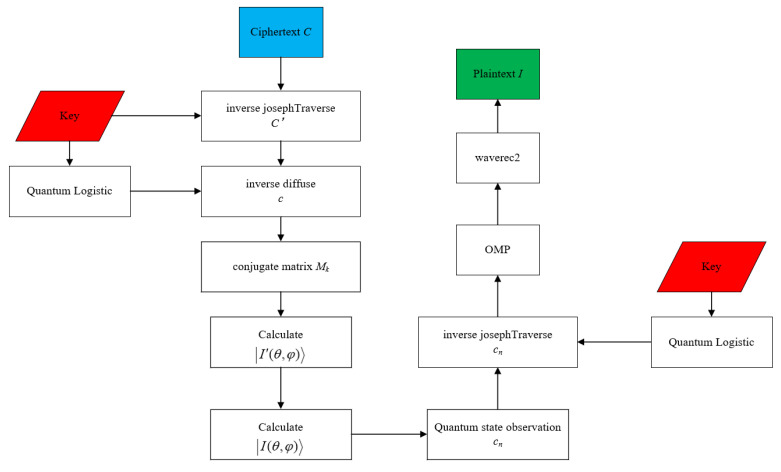
Decryption flowchart.

**Figure 6 entropy-24-00251-f006:**
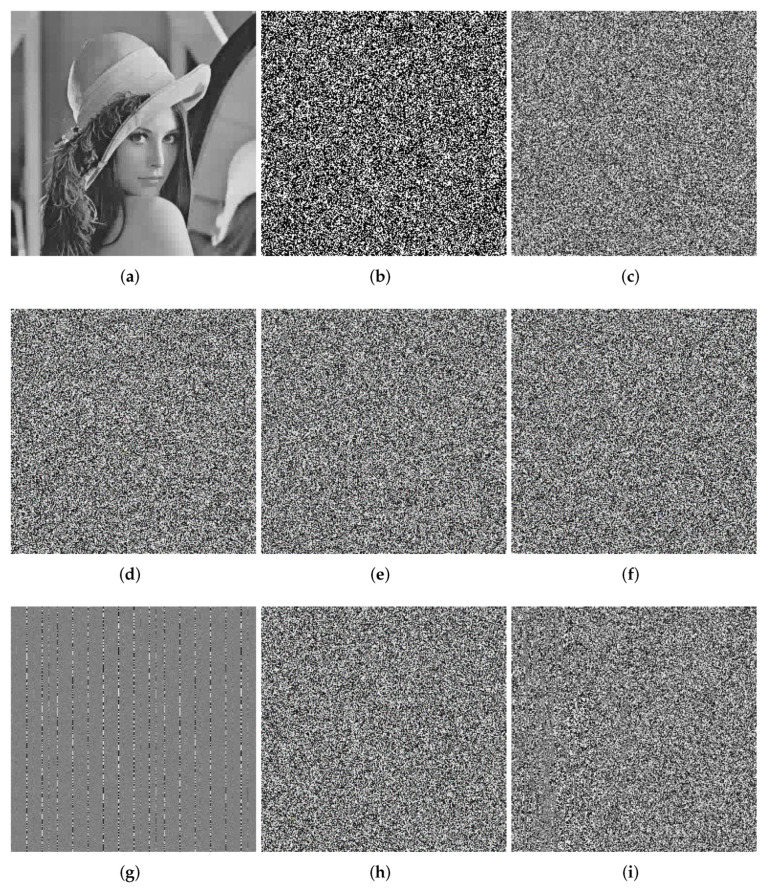
Key sensitivity analysis. (**a**) Correct key. (**b**) $x+10^{-10}$. (**c**) $y+10^{-10}$. (**d**) $z+10^{-10}$. (**e**) $r+10^{-10}$. (**f**) $\beta +10^{-10}$. (**g**) $\theta +10^{-10}$. (**h**) $q+10^{-10}$. (**i**) $index+10^{-10}$.

**Figure 7 entropy-24-00251-f007:**
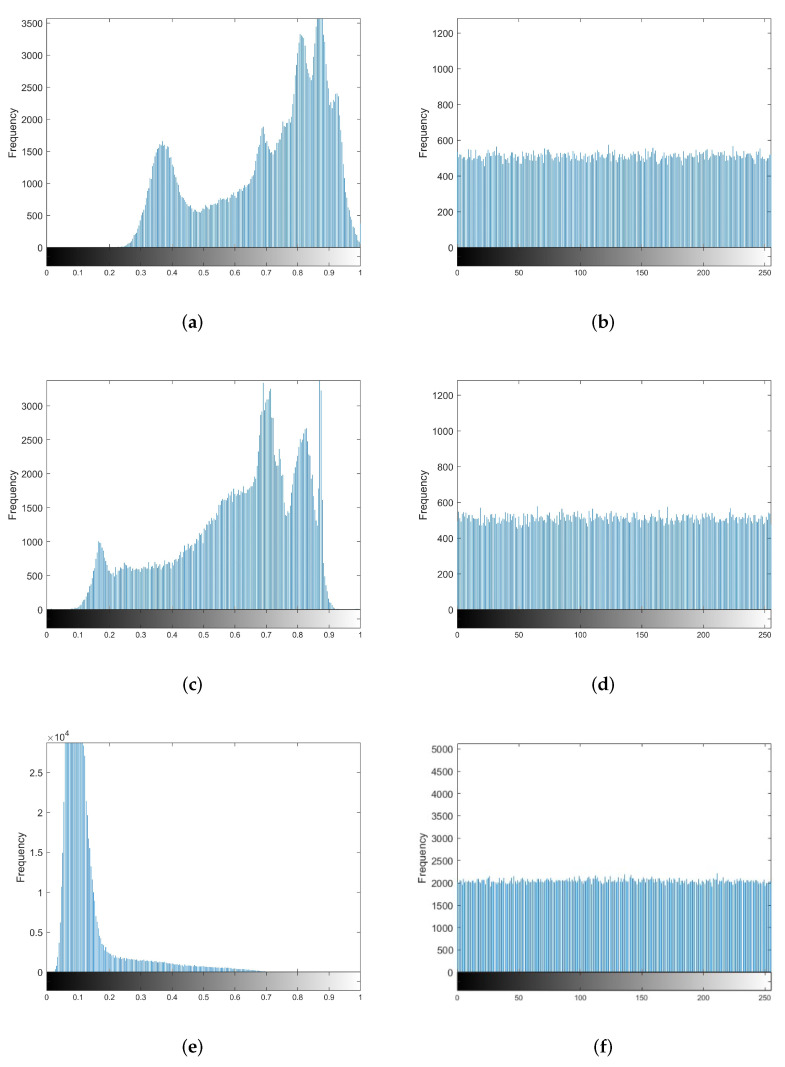
Histogram. (**a**) Lena. (**b**) Ciphertext of Lena. (**c**) Boat. (**d**) Ciphertext of Boat. (**e**) 7.2.01.tiff. (**f**) Ciphertext of 7.2.01.tiff.

**Figure 8 entropy-24-00251-f008:**
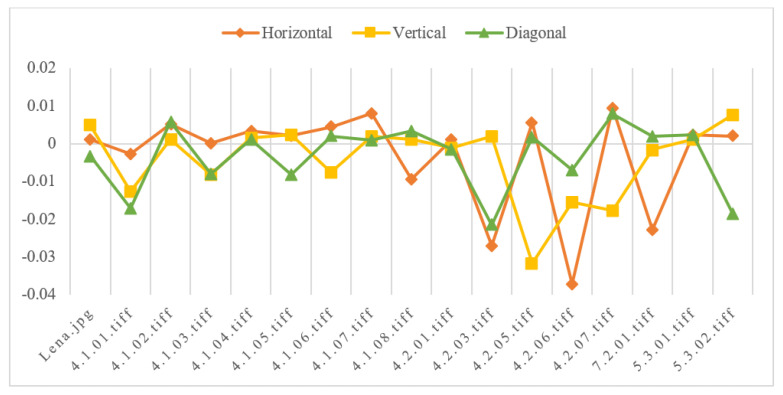
Adjacent pixel correlation of ciphertext.

**Figure 9 entropy-24-00251-f009:**
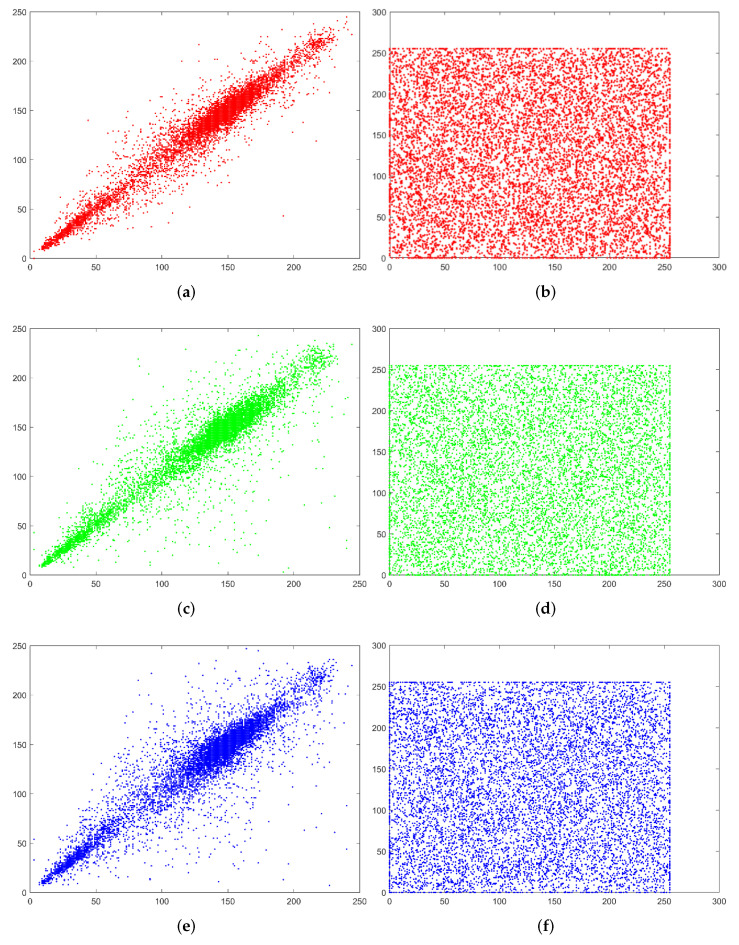
Lena’s adjacent pixel distribution. (**a**,**c**,**e**) are plaintext Lena. (**b**,**d**,**f**) are ciphertext Lena.

**Figure 10 entropy-24-00251-f010:**
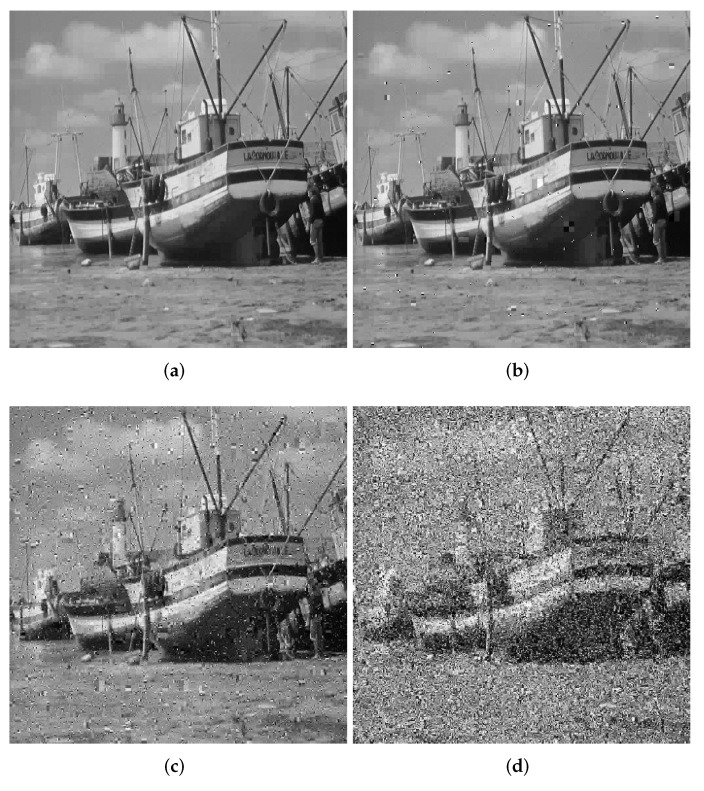
Decrypted images under different intensities of noise. (**a**) Decrypted 0.0001. (**b**) Decrypted 0.001. (**c**) Decrypted 0.01. (**d**) Decrypted 0.1.

**Figure 11 entropy-24-00251-f011:**
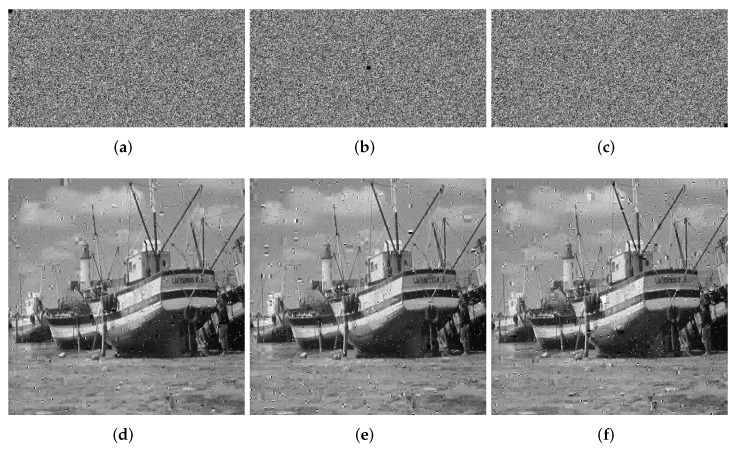
Loss data of 8 × 8. (**a**) Top left. (**b**) Center. (**c**) Bottom right. (**d**) Decrypted image (top left). (**e**) Decrypted image (center). (**f**) Decrypted image (bottom right).

**Figure 12 entropy-24-00251-f012:**
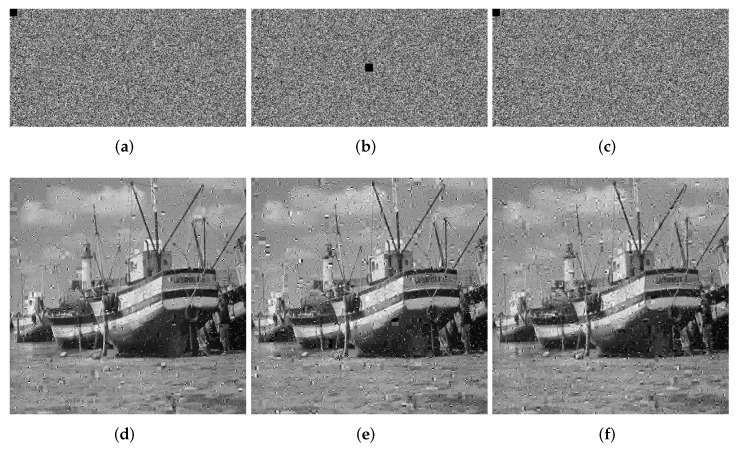
Loss data of 16 × 16. (**a**) Top left. (**b**) Center. (**c**) Bottom right. (**d**) Decrypted image (top left). (**e**) Decrypted image (center). (**f**) Decrypted image (bottom right).

**Figure 13 entropy-24-00251-f013:**
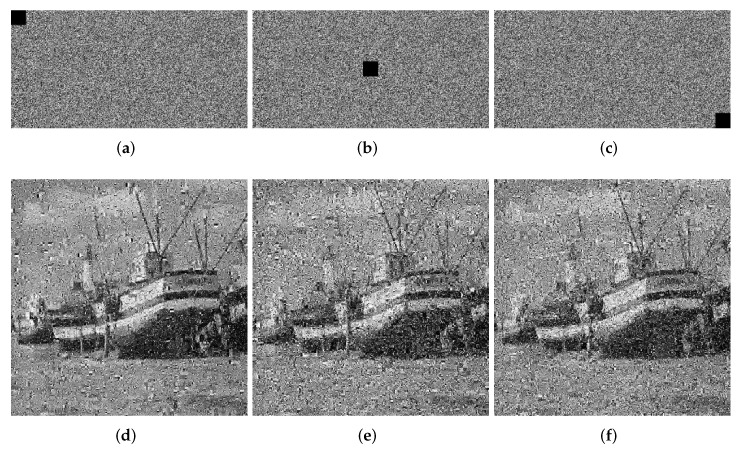
Loss data of 32 × 32. (**a**) Top left. (**b**) Center. (**c**) Bottom right. (**d**) Decrypted image (top left). (**e**) Decrypted image (center). (**f**) Decrypted image (bottom right).

**Table 1 entropy-24-00251-t001:** Keyspace analysis.

	Keyspace
**Our algorithm**	>$2^{300}$
**Reference [33]**	$2^{294}$
**Reference [4]**	$2^{250}$
**Reference [38]**	>$10^{8}$
**Reference [39]**	$2^{186}$
**Reference [40]**	$2^{149}$
**Reference [17]**	$2^{125}$

**Table 2 entropy-24-00251-t002:** Adjacent pixel correlation of ciphertext.

Size	Image	Horizontal	Vertical	Diagonal
256 × 256	Lena.jpg	0.0011	0.0050	−0.0032
	4.1.01.tiff	−0.0028	−0.0126	−0.0171
	4.1.02.tiff	0.0052	0.0011	0.0057
	4.1.03.tiff	$8.8213\times 10^{-5}$	−0.0081	−0.0079
	4.1.04.tiff	0.0034	0.0014	0.0011
	4.1.05.tiff	0.0020	0.0023	−0.0082
	4.1.06.tiff	0.0044	−0.0076	0.0020
	4.1.07.tiff	0.0080	0.0019	0.0010
	4.1.08.tiff	−0.0094	0.0011	0.0033
512 × 512	4.2.01.tiff	0.0011	−0.0010	−0.0015
	4.2.03.tiff	−0.0269	0.0019	−0.0214
	4.2.05.tiff	0.0055	−0.0317	0.0017
	4.2.06.tiff	−0.0372	−0.0155	−0.0069
	4.2.07.tiff	0.0093	−0.0177	0.0078
1024 × 1024	7.2.01.tiff	−0.0227	−0.0015	0.0018
	5.3.01.tiff	0.0022	0.0011	0.0022
	5.3.02.tiff	0.0020	0.0075	−0.0185

**Table 3 entropy-24-00251-t003:** Compare with another algorithm.

	Horizontal	Vertical	Diagonal
Our algorithm	0.0011	0.0050	−0.0032
Reference [4]	0.0062	−0.1017	0.0052
Reference [38]	−0.0072	0.0086	−0.0028
Reference [39]	0.0010	−0.0008	0.0008
Reference [40]	0.0018	0.0014	0.0034
Reference [17]	0.0009	0.0020	−0.0016

**Table 4 entropy-24-00251-t004:** Entropy.

Size	Test Image	Entropy
256 × 256	Lena.jpg	7.999382215618442
	4.1.01.tiff	7.999232866133528
	4.1.02.tiff	7.999842999290084
	4.1.03.tiff	7.999372347262450
	4.1.04.tiff	7.999533419091807
	4.1.05.tiff	7.998158601569759
	4.1.06.tiff	7.997645074489912
	4.1.07.tiff	7.998344070593309
	4.1.08.tiff	7.999008694956089
512 × 512	4.2.01.tiff	7.999808123472354
	4.2.03.tiff	7.998547327068983
	4.2.05.tiff	7.998283067128929
	4.2.06.tiff	7.998567490111828
	4.2.07.tiff	7.998694994664158
1024 × 1024	7.2.01.tiff	7.999662345147449
	5.3.01.tiff	7.999678856272011
	5.3.02.tiff	7.999649730480633

**Table 5 entropy-24-00251-t005:** Compare with another algorithm.

	Entropy
Our algorithm	7.9993
Reference [33]	-
Reference [4]	7.9968
Reference [38]	7.9985
Reference [39]	7.9977
Reference [40]	7.9986
Reference [17]	7.9993

**Table 6 entropy-24-00251-t006:** NPCR and UACI.

	Size	NPCR	UACI
256 × 256	Lena.jpg	0.996376037597656	0.336031279919194
	4.1.01.tiff	0.996520996093750	0.335112089269301
	4.1.02.tiff	0.989776611328125	0.335277267156863
	4.1.03.tiff	0.997070312500000	0.342420630361520
	4.1.04.tiff	0.996459960937500	0.336228314568015
	4.1.05.tiff	0.998565673828125	0.344157978132659
	4.1.06.tiff	0.979400634765625	0.343144435508578
	4.1.07.tiff	0.996490478515625	0.336412018420650
	4.1.08.tiff	0.987884521484375	0.336220894607843
512 × 512	4.2.01.tiff	0.996910095214844	0.335317933325674
	4.2.03.tiff	0.996177673339844	0.335232107125077
	4.2.05.tiff	0.997024536132813	0.331339602002911
	4.2.06.tiff	0.995491027832031	0.336372472426471
	4.2.07.tiff	0.996109008789063	0.337656806497013
1024 × 1024	7.2.01.tiff	0.994831085205078	0.333878127453374
	5.3.01.tiff	0.997009277343750	0.352036449955959
	5.3.02.tiff	0.996718551635742	0.336304112752279

**Table 7 entropy-24-00251-t007:** Compare with another algorithm.

	NPCR (%)	UACI (%)
Our algorithm	99.637	33.60
Reference [33]	-	-
Reference [4]	-	-
Reference [38]	-	-
Reference [39]	99.621	33.51
Reference [40]	-	-
Reference [17]	99.607	33.51

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
