# Peer review of "An Image Encryption Scheme Based on Logistic Quantum Chaos"

_entropy, 2022, doi:10.3390/e24020251_

Round 1

Reviewer 1 Report

The authors have conceived a quantum imaging algorithm. This algorithm is based on logistic quantum chaos. The authors state that this technique is stable against the standard attacks. The subject is interesting; however, I have some remarks that should be considered in the revised version:

-The author should introduce the quantum imaging and give some examples of quantum imaging algorithm such as in (Applied Sciences 10 , 4040 (2020)).

- The authors should discuss the other closer techniques and give motivations for their method.

- The authors should introduce the quantum algorithm for the readers and should include some recent ones such as ( Applied Sciences 9 (7), 1277 (2019)).

- The author is studying a closed system, what happens if we consider the dissipations which induced fluctuations and sometimes give opportunity for the hackers to use them in their advantage…. A discussion for the section on the open quantum system and different concepts should be briefly discussed, reference such as PRA 93 (4), 042116 (2016) and references therein

Reviewer 2 Report

After carefully reading the manuscript by Y. Wang et al., entitled: "An image encryption scheme based on Logistic quantum chaos," I regret to inform that I cannot recommend it, in its present form, for publication in Entropy.

The main reason for my decision is that the manuscript does not provide a realistic analysis of the proposed image-encryption method. Moreover, although the manuscript claims to provide an encryption method for the "quantum era" it is not clearly explained how quantum platforms are meant to be used with the proposed algorithm:

- Is it possible to actually implement the algorithm in currently-available quantum platforms?

- The encoded image is meant to be sent to a receiver, how is this process done? If the platform where the image is encoded is not compatible with the entities that will carry the information (probably photons), will the transduction process affect the encrypted information?

- Is the encoding process fast enough to work within the coherence time of the quantum platform?

Regarding the robustness analysis, there are some points that need to be addressed:

1) Line 231: "The authors state: "We believe that the attacker may not obtain the corresponding plaintext picture by analyzing the histogram of the ciphertext image." It is my considered opinion that this expression is not scientifically valid. Please clarify (elaborate on) whether the attacker can or cannot recover the original-image information. It seems to me that brute-force algorithms could eventually find the information. Is the parameter-space, in the histogram, sufficiently large to prevent any brute-force algorithm to find the original information?

2) Section 4.4.2: The authors crop the images using squares of 8x8, 16x16 and 32x32 pixels. I would like the authors to clarify why they choose the locations on the corners and the center. Why not trying random locations of many squares? In realistic scenarios, one would not expect to lose information in those precise locations.  

Finally, some minor remarks:

- Line 3: "Hadamm" should be "Hadamard"

- Line 84: "security" should be "securily"

- Third line after line 120: "The Goggin uses.." would read better if modified to "The Goggin method uses..."

Round 2

Reviewer 1 Report

The revised version is ok in general, however, the authors should mention that they did not include fluctuations and dissipation in their study.

Reviewer 2 Report

The authors did not include any substantial information regarding my previous reports, which is why I cannot make further assessment of the manuscript. I strongly believe that authors have to make an effort and include a discussion on the experimental feasibility of their proposal. What is the best platform to implement their algorithm? What are the typical processing times and how do they compare coherence time of the selected platform? How are they going to manage losses due to communication between emitter and receiver?

Regarding the crop-image process, although the positions of the squares were randomly selected by the computer, they do not look random at all. I still believe that an analysis with one or more than one square positioned at other locations is important, not only for proving the generality of the protocol, but also to show the reader that, in fact, the position of the squares can be randomly selected.